# Acetylcholine modulates the temporal dynamics of human theta oscillations during memory

Tamara Gedankien [1], Ryan Joseph Tan[2], Salman Ehtesham Qasim [3], Haley Moore[2], David McDonagh[4], Joshua Jacobs [1,5,6] ✉ & Bradley Lega [2,6] ✉

The cholinergic system is essential for memory. While degradation of cholinergic pathways characterizes memory-related disorders such as Alzheimer's disease, the neurophysiological mechanisms linking the cholinergic system to human memory remain unknown. Here, combining intracranial brain recordings with pharmacological manipulation, we describe the neurophysiological effects of a cholinergic blocker, scopolamine, on the human hippocampal formation during episodic memory. We found that the memory impairment caused by scopolamine was coupled to disruptions of both the amplitude and phase alignment of theta oscillations (2–10 Hz) during encoding. Across individuals, the severity of theta phase disruption correlated with the magnitude of memory impairment. Further, cholinergic blockade disrupted connectivity within the hippocampal formation. Our results indicate that cholinergic circuits support memory by coordinating the temporal dynamics of theta oscillations across the hippocampal formation. These findings expand our mechanistic understanding of the neurophysiology of human memory and offer insights into potential treatments for memory-related disorders.

Cholinergic pathways, which widely innervate the hippocampal formation, play a critical role in memory. Extensive research has shown that experimentally induced cholinergic blockade disrupts memory in both animals[1–3] and humans[4–6]. Further, patients with memory-related disorders such as Alzheimer's disease (AD) exhibit disruptions in cholinergic pathways[7,8], including brain-wide decreased density of cholinergic innervation[9]. The effectiveness of cholinesterase inhibitors, which promote cholinergic function and help attenuate memory deficits in patients with AD, further underscores the importance of cholinergic circuits to human memory and the pathophysiology of dementia[10,11]. However, the specific neurophysiological mechanisms by which acetylcholine supports memory and cognition remain poorly understood in humans. Our limited knowledge creates an obstacle to the development of new therapeutic

strategies for treating AD and the broader range of brain disorders related to cholinergic dysfunction[12–14].

An established strategy for investigating the functions of cholinergic circuits involves measuring physiological changes following the administration of cholinergic antagonists, such as scopolamine. Scopolamine, which blocks muscarinic acetylcholine receptors, strongly impairs learning during spatial and visual memory tasks in animals[15–18]. In animals, administration of cholinergic antagonists disrupts the generation and tuning of hippocampal theta oscillations[19–22] whereas cholinergic agonists induce theta activity[23,24]. Specifically, in animals, cholinergic blockade inhibits slow theta oscillations that normally appear during rest[19] as well as other behaviors[25]. However, this blockade spares movement-related fast theta, thus revealing a distinction between cholinergic-sensitive slow theta ("Type 2") and cholinergic-

[1]Department of Biomedical Engineering, Columbia University, New York, NY 10027, USA. [2]Department of Neurological Surgery, University of Texas Southwestern, Dallas, TX 75390, USA. [3]Department of Psychiatry, Icahn School of Medicine at Mount Sinai, New York, NY 10029, USA. [4]Department of Anesthesiology, University of Texas Southwestern, Dallas, TX 75390, USA. [5]Department of Neurological Surgery, Columbia University, New York, NY 10032, USA. [6]These authors contributed equally: Joshua Jacobs, Bradley Lega. ✉e-mail: joshua.jacobs@columbia.edu; bradley.lega@utsouthwestern.edu

resistant fast theta ("Type 1")[19]. This link between cholinergic modulation and theta oscillations is important because extensive research has shown that hippocampal theta oscillations contribute critically to memory processing[26–29], specifically by coordinating spike timing[30–32]. Together, these results suggest that a mechanism by which cholinergic antagonists disrupt mnemonic behavior in animals is by impairing theta oscillations and associated neural processes[33].

In humans, the main behavioral effect of scopolamine is a disruption in the ability to encode new episodic memories[5,34,35]. Importantly, the effects of scopolamine are specific to memory encoding and not retrieval, as there is minimal impact for recalling items learned prior to drug administration[6,17]. A separate line of research in humans shows that theta oscillations are critical for memory encoding, supporting the notion that the memory impairment from cholinergic blockade in humans could also result from changes in theta. Theta power in the human hippocampus reliably increases for successful episodic memory encoding[36,37]. Further, previous studies show that inter-trial phase locking in the theta band predicts successful memory[38–43], and that theta networks synchronize during memory processing[44–47]. Computational models and associated experiments further suggest that acetylcholine promotes the shifting of hippocampal theta networks between encoding and retrieval states that preferentially occur at different phases of theta[33,48,49]. Together, these findings support a hypothesis by which the cholinergic system regulates theta power and phase dynamics that underlie successful memory encoding.

Administering drugs that modulate cholinergic processes offers a direct intervention to understand the physiological effects of cholinergic tone in the human brain and examine its potential link to theta oscillations. Here, combining intracranial brain recordings, pharmacological manipulation, and behavioral experiments, we provide the first account of the neurophysiological effects of an anticholinergic drug in the human hippocampal formation during episodic memory. We administered a single dose of scopolamine (or saline, in a placebo condition) to twelve epilepsy patients prior to a verbal episodic memory task. Based on animal models[19,25], we hypothesized that scopolamine would disrupt hippocampal theta oscillations in humans, including both slow (2–4 Hz) and fast (4–10 Hz) theta bands. As such, we analyzed three physiological phenomena linked to theta oscillations and memory in humans: oscillatory power, phase reset, and synchrony. We found that scopolamine impaired memory for every subject and this impairment was accompanied by a selective disruption in slow theta power during encoding. We also found that scopolamine disrupted the phase reset of theta oscillations at the time of encoding, and that the magnitude of this disruption correlated with memory impairment. Lastly, we demonstrated that scopolamine disrupted theta-band synchrony within the hippocampal formation. These findings demonstrate that cholinergic blockade significantly influences oscillatory dynamics of theta power and phase in the hippocampal formation, suggesting potential strategies to restore memory in diseased states via selective modulation of theta oscillations.

## Results

### Scopolamine impairs episodic memory encoding

To investigate the neurophysiological effects of cholinergic modulation on memory, we asked twelve patients with surgically implanted electrodes to perform a verbal episodic memory task after double-blind administration of a cholinergic blocker, scopolamine, or a placebo (saline). The subjects in these experiments were neurosurgical epilepsy patients who had intracranial electroencephalographic (iEEG) electrodes implanted throughout their brain for seizure mapping. The electrode coverage was extensive across the hippocampal formation, including bilateral anterior and posterior hippocampus, and entorhinal cortex (Table S1). We recorded iEEG signals during both placebo and scopolamine sessions. In each trial of the episodic memory task, subjects learned a list of words and then, after a short math distractor,

tried to recall as many words as possible (Fig. 1a). To assess how the disruption of cholinergic processes affects cognition, we compared subjects' performance during free recall and the math distractor task between the scopolamine and placebo conditions.

Subjects' memory performance significantly decreased in the presence of scopolamine. Every subject individually showed a significantly lower recall probability in the scopolamine condition compared to placebo (all $p$'s < 0.05, $\chi^2$ tests), with the magnitude of this decrease ranging from ~20 to 100% (Fig. 1c). At the group level, the disrupted memory performance from scopolamine was significant, with subjects remembering an average of $31.2 \pm 3.27\%$ of items with saline, and only $10.3 \pm 3.56\%$ with scopolamine (mean $\pm$ SEM; $t(12) = 6.19$, $p = 6.79 \times 10^{-5}$, paired $t$-test) (Fig. 1b). Across subjects, the magnitude of the scopolamine-related memory disruption was not explained by variations in patients' weight (Fig. S1b), age ($r = 0.101$, $p = 0.754$, two-sided Pearson's correlation), or the interaction between weight and age ($p > 0.05$, linear mixed effect (LME) model). The effects of scopolamine were sustained throughout the experiment, as there were no significant changes in mean recall probability across trials within sessions (Fig. S2c). These results, showing the robust disruption of human episodic memory encoding following scopolamine administration, are consistent with previous experiments[5,34,35].

To better understand how scopolamine impacted the mechanisms of memory encoding, we examined whether it affected memory for item order during recall[50]. We found that, while memory was universally altered by scopolamine, there were no differences in the temporal dynamics of memory performance, such as the serial position of recalled items and the recall order (Fig. 1e–g, two-way ANOVA tests). There was also no significant difference in the rate of subjects recalling items from previous lists (list intrusions, Fig. S2b). These results are consistent with notion that scopolamine specifically affects encoding-related processes[6,17], insofar as when encoding succeeded, recall dynamics were unaffected.

We found additional evidence that scopolamine selectively affects episodic memory encoding by analyzing subjects' performance in the math distractor task. At the end of each word list, subjects performed a distractor task in which they solved a series of math problems. Notably, performance on this math task did not significantly change due to scopolamine. Accuracy on the math task was consistently high in both scopolamine and control conditions (placebo: $91.9 \pm 8.0\%$, scopolamine: $91.6 \pm 7.4\%$, $t(12) = 0.42$, $p = 0.68$, paired $t$-test) (Fig. 1d). Consistent with earlier work[51], reaction times were slower with scopolamine ($t(12) = -3.04$, $p = 0.01$, paired $t$-test) (Fig. S2a). Overall, these results indicate that the cognitive effects of scopolamine are selective, most strongly disrupting episodic memory encoding.

### Scopolamine disrupts memory-evoked slow theta oscillations and elicits broadband neural activity

We next examined the power of neuronal oscillations during memory encoding in the presence of scopolamine, building off work showing that theta oscillations in the human temporal lobe exhibit increased power during successful memory encoding[36,37,42,52]. We used spectral analysis (see Methods) to examine the power of neuronal oscillations during item presentation for all electrodes across the hippocampal formation, including contacts in hippocampus and entorhinal cortex (EC)[53]. For each electrode, we measured power at frequencies between 2 and 32 Hz, and normalized the measured power relative to the mean and standard deviation of the baseline period 500 ms before word onset at each session, electrode, and frequency (Fig. 2a–c). We then compared encoding-related shifts in theta power between scopolamine and control sessions.

As expected from earlier work[36,37,42,52], in the placebo condition theta oscillations across the hippocampal formation increased power during memory encoding. However, following scopolamine administration, this power increase was significantly lower (Fig. 2a–c).

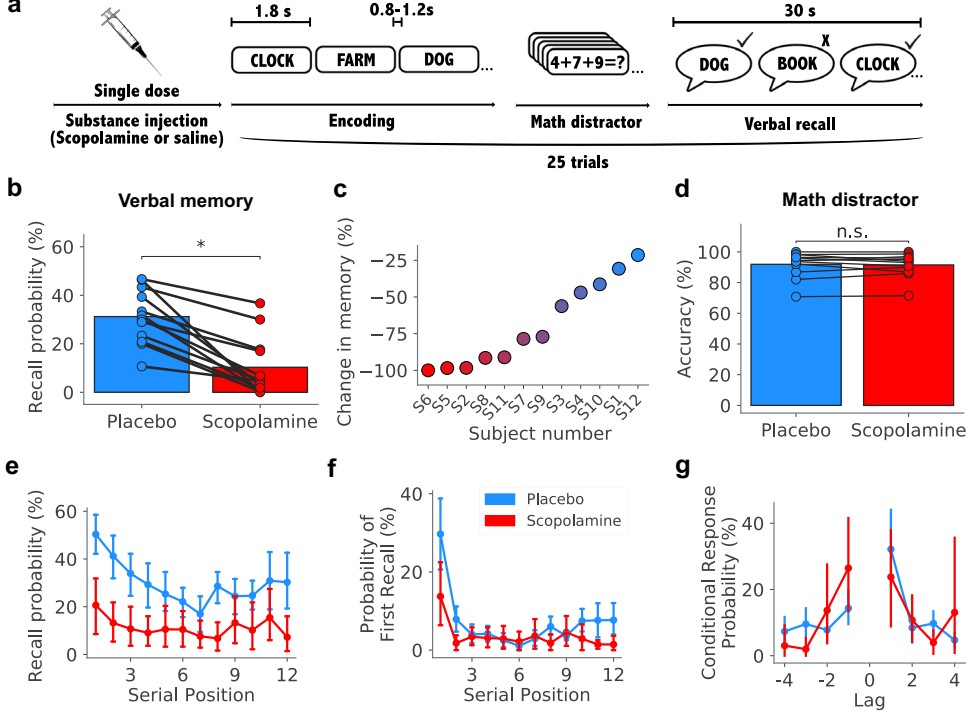

**Fig. 1 | Scopolamine impairs episodic memory encoding. a** Diagram of the task structure: prior to the task, subjects received a single dose of scopolamine (or saline, in a placebo condition). During the encoding period of the task, subjects are presented with a sequence of twelve words. Then, after completing a math distractor consisting of simple algebra equations, subjects are asked to verbally recall as many words as possible from the encoding period. **b** Bar plot showing differences in recall probability between the placebo and scopolamine sessions for all subjects ($t(12) = 6.19$, $p = 6.79 \times 10^{-5}$, two-sided paired $t$-test). **c** Percent decrease in recall probability with respect to the placebo session for all subjects. **d** Bar plot showing differences in accuracy in the math distractor between the placebo and scopolamine sessions ($t(12) = 0.42$, $p = 0.68$, two-sided paired $t$-test). **e** Recall probability by word serial position during encoding. Data are presented as mean values across subjects, and error bars indicate the 95% confidence interval. **f** Probability of first recall (PFR) by word serial position during encoding. Data are presented as mean values across subjects, and error bars indicate the 95% confidence interval. **g** Conditional response probability (CRP). Data are presented as mean values across subjects, and error bars indicate the 95% confidence interval. Source data are provided as a source data file.

To assess the robustness of this pattern, we implemented a LME model that tested for differences in the time-averaged power during encoding for slow (2–4 Hz) and fast (4–10 Hz) theta frequency bands separately. We found that power was significantly lower during scopolamine compared to placebo, and this effect was restricted to the 2–4-Hz slow theta band (Fig. 2c–d, Fig. S3). The effect was strongest in the posterior hippocampus (PH: $t(23) = -2.44$, $p = 0.015$, LME model, multiple-comparison corrected) (Fig. 2d), and it was present when including all encoding trials as well as forgotten trials only (Fig. S4). Notably, this 2–4-Hz band is the same range where theta power was linked to subsequent memory effects (SME) in earlier human studies examining episodic memory encoding[37,42,52,54], and it is also the same band where we found a trend towards greater power for remembered than forgotten trials within the placebo session (Fig. S5). We considered the possibility that the magnitude of the disruption in theta oscillations related to scopolamine concentration or the magnitude of memory impairment, but there was no significant correlation with either concentration ($r = 0.091$, $p = 0.790$, two-sided Pearson's correlation) or memory (Fig. S6a).

In addition to examining the effect of scopolamine on narrowband oscillations, we measured broadband power, which captures aperiodic activity in the brain[55]. Whereas narrowband theta oscillations reflect synchronous population events that are adaptive to behavioral states, aperiodic activity reveals the balance between excitation and inhibition in local neural circuits[56]. We measured broadband power (2–128 Hz) during memory encoding[57] (see Methods), and compared the resulting magnitude of broadband power (or the 1/f power spectrum's "offset") between the scopolamine and placebo conditions. Across all trials, broadband power was significantly higher in the

scopolamine condition compared to placebo. An example of this effect can be seen in Fig. 2e, where the measured power for one electrode was greater at all frequencies for scopolamine compared to placebo. Because this effect was present at all frequencies, the difference primarily reflects a broadband pattern rather than narrowband oscillations. This effect was robustly present at the population level as demonstrated by the group-level distribution of power spectral density (PSD) offset values (Fig. 2f), and the high percentage of electrodes showing significantly greater power at all frequencies in the presence of scopolamine compared to placebo (Fig. 2g, all $p$'s < 0.05, Wilcoxon rank-sum tests).

These results indicate that scopolamine changes the power of iEEG activity during memory processing by disrupting encoding-related narrowband slow theta oscillations within a background of increased broadband activity. These results are consistent with a model by which cholinergic blockade leads to a disruption in rhythmicity among neuronal populations, thus resulting in a decrease in the amplitude of narrowband theta oscillations and a simultaneous increase in aperiodic neuronal excitation.

### Scopolamine disrupts the phase reset of theta oscillations
Drawing upon evidence that the phase of ongoing neuronal oscillations supports memory processes[39,48,58–60], we examined how scopolamine modulated the phase dynamics of theta oscillations and whether those changes were linked to behavior. First, to ensure we were measuring robust theta oscillations, we implemented an established oscillation-detection procedure[61] (see Methods). Next, we computed the inter-trial phase coherence (ITPC) during memory encoding to test for phase reset of theta oscillations following stimulus

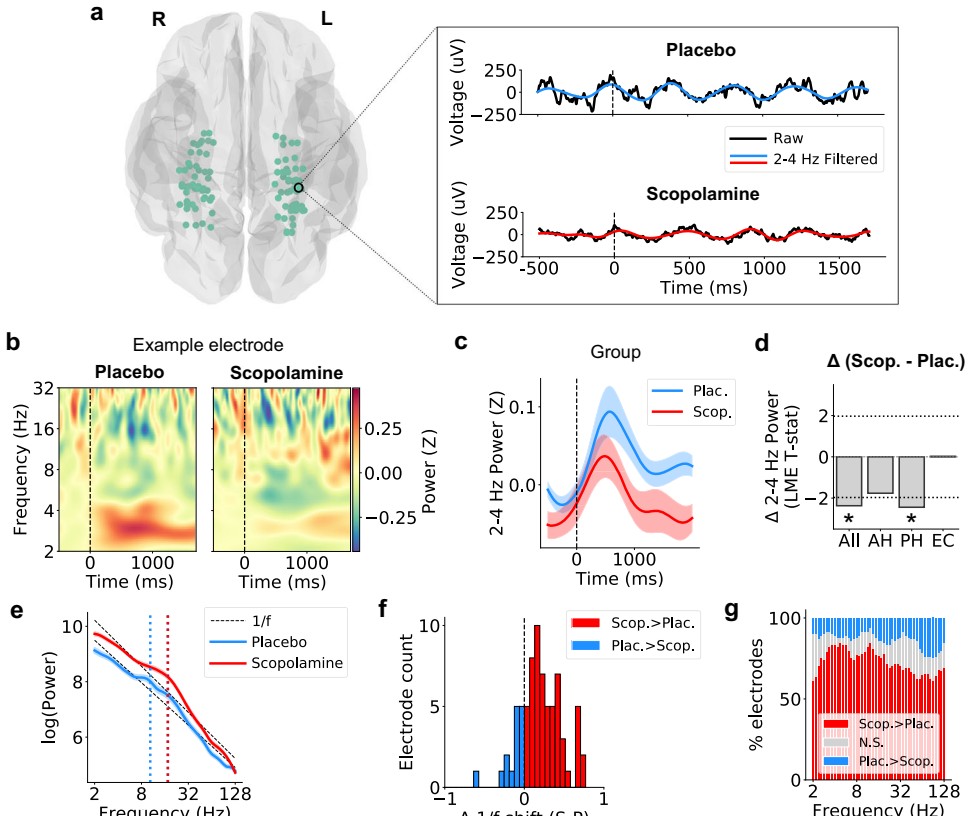

**Fig. 2 | Scopolamine disrupts memory-evoked slow theta oscillations and elicits broadband neural activity. a** Left: anatomical locations of all electrodes analyzed in this study. Brain plot was created using the nilearn software package. Copyright (c) 2007–2023 The nilearn developers. All rights reserved. Right: example trials from a hippocampal electrode from subject 5, showing onset of slow theta oscillations following encoding cue ($t = 0$). **b** Time-frequency spectrogram showing memory-evoked normalized power for the same example electrode across all trials. The increase in 2–4 Hz slow theta power seen during the placebo session is not present during the scopolamine session. **c** Group-level mean normalized power for the placebo (blue) and scopolamine (red) conditions. Shading denotes ± SEM. **d** Plot showing statistical differences in time-averaged 2–4 Hz normalized power between placebo and scopolamine for electrodes within all hippocampal formation subregions combined (All), anterior hippocampus (AH) only, posterior hippocampus (PH) only, and entorhinal cortex (EC) only. Dashed lines indicate 95% confidence intervals. Asterisks denote statistical significance. The effect was strongest in the PH ($t(23) = -2.44$, $p = 0.015$, LME model, multiple-comparison corrected). **e** Power spectral density (PSD) for an example hippocampal electrode from subject 2 during encoding. Shading denotes ± SEM. **f** Distribution of 1/f offset change values (scopolamine−placebo) across all electrodes. **g** Percentage of electrodes showing significant increases in power for each condition at each frequency (all $p$'s < 0.05, two-sided Wilcoxon rank-sum tests). Source data are provided as a source data file.

presentation[62,63]. For a given electrode, the ITPC quantifies the consistency of the phase alignment of neuronal oscillations at particular timepoints across trials (Fig. 3a). We found that theta oscillations in the placebo condition showed significant phase reset following stimulus presentation, although individual electrodes reset to different phases ($p > 0.05$, Rayleigh tests) (Fig. S7). However, in the scopolamine condition, the magnitude of theta phase reset was significantly lower. This disruption of theta phase reset by scopolamine can be seen robustly in signals from individual electrodes (Fig. 3b, Fig. S8) as well as at the group level (Fig. 3c) ($n = 69$, $p < 0.05$, cluster permutation test).

Because the magnitude of memory impairment varied across subjects (Fig. 1c), we considered the possibility that changes in phase reset vary with intersubject differences in memory performance. To test this hypothesis, we computed the correlation between the mean change in phase reset and changes in memory performance caused by scopolamine across subjects. We found a significant correlation between the magnitude of the change in theta-band phase reset and the severity of memory impairment ($r = 0.739$, $p = 0.015$, two-sided Pearson's correlation) (Fig. 4b). Thus, subjects who showed more severe memory impairments from scopolamine exhibited a greater disruption of theta phase reset. This effect was stronger in the fast theta band (4–10 Hz) (Fig. 4b), but it was also present in the slow theta band (2–4 Hz) ($r = 0.658$, $p = 0.039$, two-sided Pearson's correlation) (Fig. S9).

We considered the possibility that scopolamine's apparent disruption of theta power and phase reset reflected a disruption of an evoked signal or changes in power rather than the phase of an ongoing oscillation. To evaluate these possibilities, we first examined whether changes in event-related potentials (ERPs) were related to changes in power[63]. We did not find significant correspondence between the magnitude of power changes and ERP differences (Fig. S10a, b), and we also did not find a significant correlation between peak ERP amplitude changes and time-matched power changes ($r = -0.117$, $p = 0.330$, two-sided Pearson's correlation) (Fig. S10c). Also, changes in phase reset did not significantly correlate with changes in either slow or fast theta power ($p$'s > 0.4, Pearson's correlations) (Fig. S11). Consistent with suggested best practices for these analyses[63,64], these results indicate that the observed phase reset cannot be explained by changes in ERPs or power, and instead more directly reflect changes in the dynamics of theta phase.

**Scopolamine disrupts phase synchrony in the theta band**
Previous studies showed that theta oscillations synchronize neuronal activity across brain regions during memory processing[44,45,65,66]. We therefore tested whether scopolamine impacted phase synchronization within the hippocampal formation during memory encoding.

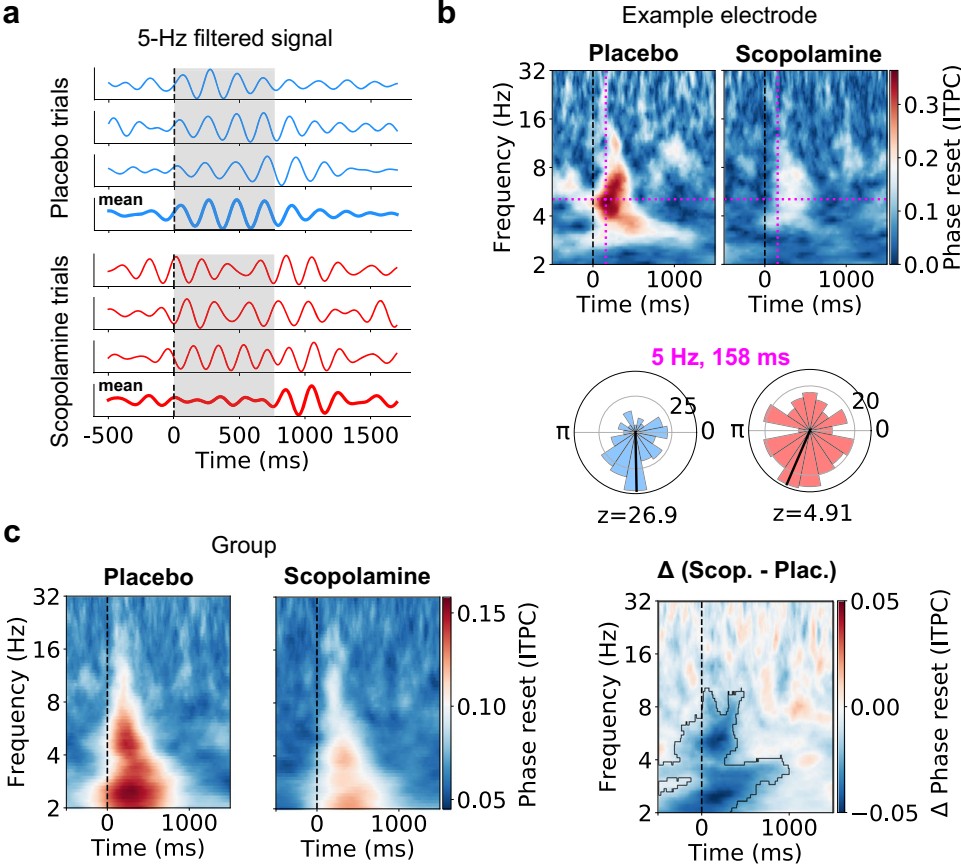

**Fig. 3 | Scopolamine disrupts the phase reset of theta oscillations. a** 5-Hz-filtered EEG from an example hippocampal electrode from subject 12 for selected trials. Unlike the placebo condition, the 5-Hz-filtered oscillations in the scopolamine condition do not show phase alignment across trials. **b** Top: Inter-trial phase coherence (ITPC, or phase reset) during encoding for the same example electrode for the placebo and scopolamine conditions (0 ms denotes time of encoding cue). The increase in 5-Hz theta phase reset seen during the placebo session is not present during the scopolamine session. Magenta dashed lines indicate peak in ITPC between 0–300 ms for the placebo condition. Bottom: circular histograms showing phase distributions at the peak in ITPC following encoding cue for placebo (blue) and scopolamine (red). **c** Left: group-level mean phase reset across all hippocampal formation electrodes for the placebo and scopolamine conditions. Right: group-level mean change in phase reset between placebo and scopolamine. Black outline indicates significant clusters of change in phase reset ($n = 69$, $p < 0.05$, cluster-based permutation test, multiple-comparison corrected). Source data are provided as a source data file.

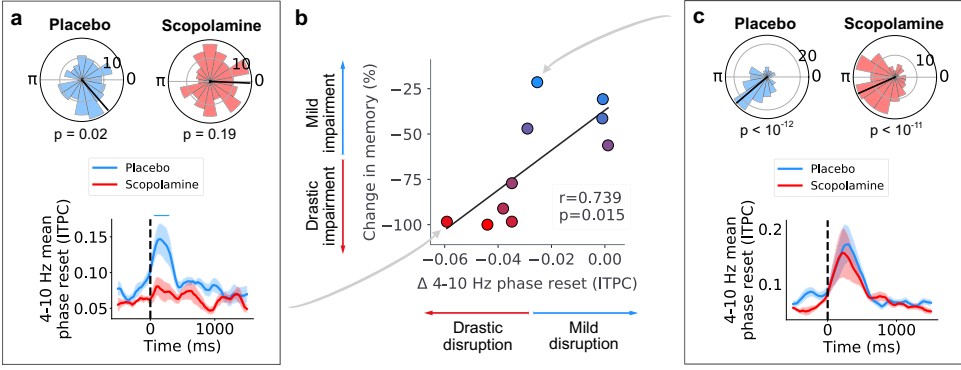

**Fig. 4 | Theta phase reset disruption correlates with memory impairment. a** Top: circular histograms from an example hippocampal electrode from subject 5, displaying significant preferred phase during placebo but not during scopolamine conditions ($p = 0.02$ and $p = 0.19$, respectively, Rayleigh tests). Bottom: line plot showing mean phase reset across all electrodes for subject 5. Shading denotes ± SEM. Bar denotes statistical significance ($p < 0.05$, two-sided Wilcoxon rank-sum test). **b** Correlation between subjects' mean phase reset changes in fast theta during encoding and subjects' percent change in memory ($r = 0.739$, $p = 0.015$, two-sided Pearson's correlation). Subjects with greater memory impairment following scopolamine showed greater disruption in phase reset. **c** Top: circular histograms from an example entorhinal cortex (EC) electrode from subject 1, displaying significant preferred phase during placebo and scopolamine conditions ($p < 10^{-12}$ and $p < 10^{-11}$, respectively, Rayleigh tests). Bottom: line plot showing mean phase reset across all electrodes for subject 1. Shading denotes ± SEM. Source data are provided as a source data file.

To measure the synchrony of the activity between different electrode locations, we first measured the phase of the theta oscillations at each timepoint throughout the recordings. Next, for all pairs of electrodes within a subject's hippocampal formation, we measured oscillatory synchrony by comparing the differences in the instantaneous phase between the electrodes. If two electrodes were synchronized then their phase difference should be approximately constant over extended periods of time. Figure 5a shows two illustrative examples of this phenomenon in the data. In one sample trial from the placebo session (Fig. 5a, left panel), a pair of hippocampal electrodes showed a consistent phase difference of $\sim \frac{\pi}{2}$ rad for a 6–8 Hz-filtered theta oscillation. In a trial from the scopolamine

session (Fig. 5a, right panel), this same electrode pair showed a wide range of phase differences for a theta oscillation of the same frequency range. Thus, whereas there was strong synchrony between the signals measured across this electrode pair when the subject took the placebo, this synchrony was disrupted in the presence of scopolamine.

To quantify this phenomenon systematically, we measured the strength of phase synchrony between all electrode pairs by computing the phase locking value (PLV; see Methods) both in the placebo and scopolamine conditions[67]. The PLV provides a quantitative measure of the phase alignment between two electrodes, ranging between 0 (no alignment) and 1 (strong alignment). For each electrode pair, we tested

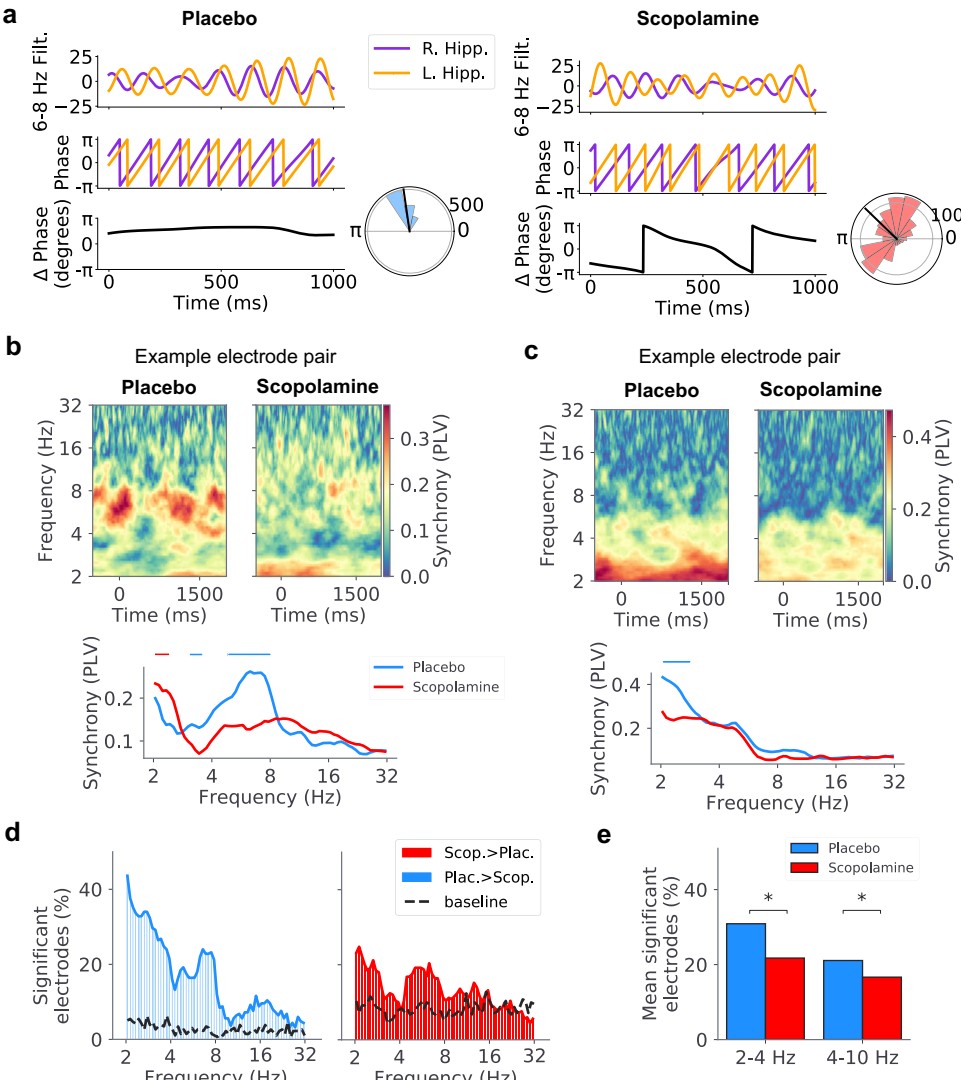

**Fig. 5 | Scopolamine disrupts phase synchrony in the theta band. a** Top-down: 6–8 Hz filtered traces, phases, phase differences, and circular histograms showing the phase differences between a right and left hippocampal electrode pair during an example trial in the placebo and scopolamine conditions for subject 4. **b** Top: Time-frequency spectrogram showing the phase-locking values (PLVs) for a hippocampal electrode pair from subject 4 during encoding for the placebo and scopolamine sessions. Warmer colors indicate greater PLV (synchrony). Bottom: Mean PLV as a function of frequency for the spectrogram shown above. Bars indicate frequency windows where statistical significance was present (blue = Plac. > Scop., red = Scop. > Plac., $p < 0.05$, two-sided Wilcoxon rank-sum test). **c** Top: Time-frequency spectrogram showing the PLVs for a hippocampal electrode pair from subject 5 during encoding for the placebo and scopolamine sessions. Warmer colors indicate greater PLV (synchrony). Bottom: Mean PLV as a function of

frequency for the spectrogram shown above. Bars indicate frequency windows where statistical significance was present (blue = Plac. > Scop., red = Scop. > Plac., $p < 0.05$, two-sided Wilcoxon rank-sum tests). **d** Percentage of electrodes showing significant increases in synchrony at each frequency for placebo (left; blue) and scopolamine (right; red) conditions ($p < 0.05$, Wilcoxon rank-sum test). Dotted lines indicate the percentage of electrodes showing significant increases in synchrony during the baseline periods for placebo (left) and scopolamine (right) (see Methods). **e** Bar plot showing differences in the percentage of electrodes with significant increases in slow (2–4 Hz) and fast (4–10 Hz) theta-band synchrony for placebo and scopolamine (slow theta: $z = -2.87$, $p = 4.06 \times 10^{-3}$, fast theta: $z = -2.09$, $p = 0.04$, two-sided two-proportion $z$-tests). Source data are provided as a source data file.

whether the mean PLV significantly shifted between placebo and scopolamine conditions.

Overall, many electrode pairs showed significant decreases in PLV following scopolamine administration, relative to placebo. This effect can be seen in different narrowband frequencies within the theta band, with the precise band often varying between subjects (e.g., 6–8 Hz in Fig. 5b, and 2–3 Hz in Fig. 5c). At the group level, slow and fast theta-band synchrony was significantly lower for the scopolamine condition compared to placebo (slow theta: $z = -2.87$, $p = 4.06 \times 10^{-3}$, fast theta: $z = -2.09$, $p = 0.04$, two-sided two-proportion $z$-tests) (Fig. 5e). This effect was driven by an increase in theta synchrony during memory encoding in the placebo condition, as this increase was not present during the non-memory baseline period (dotted lines, Fig. 5d). This relatively lower level of synchrony from scopolamine was stable throughout the memory encoding period, as we found no differences in the level of slow- or fast-theta synchrony between the first and second halves of the encoding period ($p$'s > 0.05, paired $t$-tests). Together, these results indicate that scopolamine disrupts connectivity within the hippocampal formation by impairing the phase synchrony of theta-band oscillations during encoding. Notably, this effect was present for intra- as well as inter-hemispheric electrode pairs (Fig. S12a), suggesting that cholinergic pathways have a role in coordinating the timing of hippocampal activity both within and across hemispheres.

To test whether changes in synchrony might relate to the patterns of phase reset described above, we repeated this synchrony analysis after excluding the first 500 ms following word onset, which was the interval that showed the strongest phase reset. Here, we also found decreased theta-band synchrony following scopolamine compared to placebo (all $p$'s < 0.05, two-proportion $z$-tests) (Fig. S13). Additionally, we did not find significant correlations between changes in synchrony and changes in power (all $p$'s > 0.05, two-sided Pearson's correlations) (Fig. S14) or a significant difference in the peak frequency of the oscillations present in each condition ($p > 0.05$, Wilcoxon rank-sum test) (Fig. S15), indicating that differences in power do not explain our observations of differences in phase synchrony.

Overall, these findings show that scopolamine decreases connectivity, as measured by theta phase synchrony, across the hippocampal formation. Taken together with our analyses of oscillatory power and phase reset, our findings suggest that cholinergic circuits critically underlie the generation and synchronization of theta oscillations in the hippocampal formation.

## Discussion

Despite extensive evidence of the importance of the cholinergic system for episodic memory encoding, the neurophysiological mechanisms that underlie this relationship remain poorly understood in humans. Here, using rare intracranial recordings, we described the physiological effects of cholinergic blockade on the human hippocampal formation during episodic memory. We demonstrated that scopolamine administration suppressed the amplitude of narrowband oscillations in the slow theta band (2–4 Hz) (Fig. 2), and interrupted the resetting of theta phase following stimulus presentation (Fig. 3). Across subjects, this disruption of phase reset correlated with the magnitude of memory impairment caused by scopolamine (Fig. 4). Scopolamine also disrupted the synchrony of theta oscillations across the hippocampal formation (Fig. 5). These findings suggest that one role for cholinergic processes in memory is to coordinate the temporal dynamics of neuronal activity, in particular by allowing network patterns related to theta rhythms to flexibly change their timing to match behavioral events. Because theta oscillations coordinate the timing of single-neuron spiking across the brain[30,32,68], our results suggest that scopolamine may impair memory by disrupting the theta-dependent temporal coordination of neuronal activity across the hippocampal memory network.

Our behavioral findings build on a line of studies that examined the behavioral effects of cholinergic blockers in animals[2,3,15–18,69,70] and humans[5,6,34,35]. In humans, it is well known that cholinergic blockade disrupts episodic memory encoding[5,6,34]. Our behavioral results replicate these findings, because we found that scopolamine significantly impaired verbal episodic memory encoding during free recall, without impairing performance in a non-memory control task (solving math problems). Our results also demonstrated that key features of the behavioral dynamics of recall, such as recall order, were preserved under scopolamine. These findings suggest that scopolamine primarily impairs the overall efficiency of memory encoding, as measured by the number of items remembered, rather than changing more subtle computational aspects of memory processes that may cause changes in recall dynamics.

In addition to behavioral studies in humans, research in animals has probed the physiological basis of cholinergic antagonists. The traditional view from animal models is that cholinergic antagonists ablate immobility-related slow theta oscillations ("Type 2") but spare movement-related fast theta oscillations ("Type 1")[19,25]. Our human results are consistent with these findings, as we found that during memory encoding scopolamine most strongly disrupted the amplitude of narrowband slow theta (2–4 Hz) rather than fast theta (>4 Hz) oscillations. However, we also found some differences compared to these previous reports, which suggests that there is a more complex relationship between cholinergic processes and theta in humans. Unlike animal models, where there seems to be a strict dissociation between cholinergic-sensitive slow theta and cholinergic-insensitive fast theta oscillations[19,71], in humans we found scopolamine-related phase and synchrony disruptions at a range of theta-band frequencies. The ability of scopolamine to affect oscillations at a range of theta frequencies, including power decreases at the slow theta band and shifts in phase alignment and synchrony at both slow and fast theta, suggests that there may be a complex interacting network of different theta oscillators in the human hippocampal region. These oscillators, each with their own intrinsic frequency, might interact with one another to support cognition and be affected by cholinergic blockade both directly and indirectly—thus, leading to observed effects across the broad range of theta. This model is consistent with earlier findings of theta being spread across a range of frequencies in both humans[72] and animals[73]. With this model in mind, because there is coupling between distinct theta generators, we believe that the observed changes in fast theta phase reset and synchrony could be driven indirectly by power effects in slow theta oscillators and/or by a selective effect of scopolamine on fast theta oscillators.

Computational models have proposed a number of mechanisms for how cholinergic processes support memory, including the notions that acetylcholine enhances encoding by modulating synaptic excitability and enhancing theta-band oscillations[74,75]. Consistent with these models, we found that scopolamine affected slow theta oscillations following item presentation during memory encoding—notably, this is the specific time period that showed memory-related changes in theta power in earlier studies[36,37]. The consequences of disrupting this rhythmicity, especially in populations of interneurons[76], are a diminished amplitude of theta oscillations and a simultaneous increase in aperiodic neuronal excitation. In human intracranial brain recordings, the mean level of neuronal excitation can be estimated from the mean broadband power across all frequencies[55,77]. Consistent with this model, we show that broadband power increased in the presence of scopolamine (Fig. 2e–g). These hypothesized effects of cholinergic blockade on hippocampal neuronal populations could also help explain the disrupted phase patterns we observed following scopolamine administration. Scopolamine may disrupt the ability of local oscillators to synchronize with projecting inputs[78], thus causing local oscillators that generate theta-band rhythms to function more independently with decreased coupling to neighboring networks.

This decreased oscillatory coupling may explain our finding of decreased interhemispheric and intrahippocampal theta synchrony following scopolamine, as well as our finding of decreased theta phase reset, in which hippocampal theta rhythms normally align following mnemonically-relevant inputs[79].

Theta oscillations are important for coordinating spike timing and this may explain why the theta disruption from scopolamine causes memory impairment. In animals and humans, item and context representations encoded by hippocampal neurons require coordination with local theta oscillations via phase precession[30,80] and phase locking[32,81]. Based on this literature, our results suggest that the way in which cholinergic blockade impairs memory performance is by disrupting theta-related temporal dynamics, and thus disrupting the normally organized timing of single-neuron spiking. One study in rodents is consistent with this view, showing that scopolamine in rodents disrupted hippocampal neurons' phase precession while sparing rate coding[82]. Given the role of theta phase in modulating the timing of neuronal spiking[32,81,83], we hypothesize that the reset of theta phase that occurs after stimulus presentation is important because it causes particular phase-coupled neuronal assemblies to activate at precise timepoints during stimulus processing[84]. When phase resetting is disrupted by scopolamine, it therefore impairs the optimal timing patterns of neuronal spiking that normally occur. Further, our connectivity analysis revealed that scopolamine impaired regional connectivity by disrupting theta-band synchrony within and between bilateral brain regions. Together, these findings suggest that cholinergic blockade may impair memory encoding by disrupting the precise timing of neuronal activity across the bilateral hippocampal network that is normally coordinated by theta oscillations.

Tracing experiments in rodents suggested two pathways that could link muscarinic antagonism with theta oscillatory disruption: a "direct" pathway by which cholinergic neurons in the basal forebrain (BF) modulate hippocampal interneurons, and an "indirect" pathway by which cholinergic neurons synapse locally in the BF and modulate hippocampal projecting neurons that alter rhythmicity[85]. We considered whether these models could explain the mechanisms underlying our findings by conducting a supplementary analysis where we reanalyzed a previously published human single-nucleus RNA sequencing dataset[86] to determine the expression of muscarininc (CHRM) receptors in the human hippocampus. We found that hippocampal inhibitory interneurons preferentially express CHRM3, a Gq-coupled muscarinic receptor bearing significant homology with the CHRM1 subtype identified in rodent interneurons (see Fig. S16 and Supplementary Discussion). Thus, because muscarinic receptors are present in the human hippocampus, these findings provide preliminary support for a direct pathway as a plausible mechanism for theta modulation in our data. Additional studies are needed to determine if indirect pathways also play a parallel role in modulating human theta. Similarly, additional in vitro work on human tissue samples is likely necessary to confirm that the effects of scopolamine in the human hippocampus rely overall on postsynaptic blockade of muscarinic (i.e., CHRM) receptors rather than an increase in synaptic acetylcholine due to autoreceptor blockade[87].

Since scopolamine was applied intravenously, we cannot exclude the possibility that our observations are indirectly affected by off-target effects on regions outside of the hippocampus. A known side effect of scopolamine is pupil dilation, which leads to decreased visual contrast sensitivity and impaired visual recognition[88,89], which could impair memory. However, we think it is unlikely that vision impairments generated our results because in our task words were presented in a large and legible font, and no subjects reported issues with reading and comprehension. Additionally, previous studies also show evidence for effects of cholinergic modulation on some non-mnemonic behaviors[90–94], raising the possibility that the cholinergic impact on episodic memory occurs indirectly through primary effects on non-memory-related circuits. However, because we found preservation of performance during the math control task, we believe that effects on non-mnemonic circuits would be secondary to that on memory systems. Consistent with view, a number of prior studies show that the effects of scopolamine were focused on the hippocampal formation[20–22,95] and demonstrate that scopolamine specifically impairs episodic memory encoding, which is a behavior known to be hippocampally dependent[96]. Additionally, as noted above, our supplementary analysis of gene expression data using tissue from human patients supports the direct action of scopolamine on cholinergic synapses in the hippocampus.

Our results suggest a number of important questions for future research to clarify the specific mechanisms by which acetylcholine-dependent theta oscillations support the computational processes underlying memory. For example, it will be important to determine whether acetylcholine differentially modulates theta oscillations during memory retrieval as compared to encoding, as suggested previously[97], as well as if the same neurophysiological mechanisms underlie effects on the encoding of familiar versus novel items[3]. It would also be useful to test the effects of administering cholinergic agonists (e.g., donepezil) to intracranial patients, and testing whether the memory disruption described here can be reversed, with commensurate "rescue" of theta power and synchrony. Lastly, future work should also focus on further characterization of the distinct functions and sources of slow and fast theta rhythms, with particular focus on examining how cholinergic modulation can elucidate the two-theta model at the cellular level.

In summary, our findings provide evidence that the cholinergic system is closely involved in the oscillatory dynamics of theta power and phase in the human hippocampal formation, and support a model by which cholinergic-sensitive theta oscillations support memory formation through phase and power shifts. More broadly, our findings suggest that cholinergic circuits play a role in regulating the temporal dynamics of theta-band activity and allowing it to vary with behavior to support episodic memory formation. Going forward, our results provide a roadmap for using pharmacological modulation as a tool to identify the specific physiological mechanisms of human cognition, in particular dissociating how slow and fast theta oscillations differentially support aspects of memory processing. Lastly, our findings suggest that improving the temporal coordination of neuronal activity, specifically those involving theta oscillatory dynamics, may provide a novel route to treating memory-related disorders such as AD, as well the broader range of brain diseases related to cholinergic dysfunction.

## Methods

### Participants

The study included twelve subjects who were undergoing iEEG monitoring as part of their treatment plan for medication-resistant epilepsy. The patients had intracranial depth electrodes that were surgically implanted for the purpose of seizure localization. Participants came from the University of Texas Southwestern (UTSW) epilepsy surgery program across a time span of 2 years.

### Scopolamine administration

Participants received a 0.5-mg dose of scopolamine or saline (placebo) via an intravenous line. Intravenous scopolamine starts taking effect about 5 min after administration, and has a half-life of ~69 min[98]. Subjects completed the free-recall task ~15 min following injection. A second session using the alternate agent (scopolamine or saline) took place the following day or at least four half-lives after the first session. The test administrator and participant were blinded to the drug/placebo randomization.

### Ethics and safety

The protocol was approved by the UTSW Institutional Review Board on Human Subjects Research prior to data collection, and all participants

provided informed written consent. Physiologic monitoring of blood pressure, electrocardiogram, oxygen saturation, and mental status was performed just prior to and for one hour after scopolamine injection. Intravenous scopolamine was administered by the participant's attending nurse, and a board-certified anesthesiologist was present at the time of injection and remained available throughout the duration of the experiment to treat any potential adverse reactions. Low doses of scopolamine (such as 0.5 mg) are commonly administered in clinical settings and generally do not pose major risk factors[98]. There were no reports of adverse events from substance administration in either the drug or placebo conditions.

## Memory task

Each participant completed two sessions (drug and placebo) of a delayed free-recall task. During encoding, subjects were presented with 12 words, one at a time, and later during retrieval they were asked to verbally recall as many words as possible from the list. Each word was displayed in capital letters for 1800 ms, followed by an 800–1200 ms blank interstimulus interval to decorrelate physiological responses from successive word presentations. Words were selected at random, without replacement, from a pool of nouns (http://memory. psych.upenn.edu/WordPools). Following each list presentation, subjects were given a series of simple arithmetic problems in the form A + B + C = ?, where A, B and C were random positive integers. Next, a row of asterisks accompanied by a 300-ms, 60-Hz auditory tone was presented to indicate the start of the recall period. During the 30-s recall period, subjects were instructed to recall as many words as they could (in any order) from the most recent list. Each session included 25 lists of 12 words, for a total of 300 encoded words (Fig. 1a). At the beginning of each list, there was a baseline period in which subjects stared at a 10-s countdown on the screen.

## Intracranial recordings

Patients were implanted with either Ad-Tech Medical or PMT depth electrodes with 0.8 mm in outer diameter, forming 10–14 recording contacts arrayed at 4–5 mm intervals along the shaft of the electrode (Ad-Tech: 10 electrodes with variable spacing; PMT: 10–14 electrodes with uniform spacing, depending upon the overall depth of insertion). Intracranial EEG was sampled at 1 kHz on a Nihon-Kohden 2100 clinical system under a bipolar montage with adjacent electrodes as reference. Electrodes from hemispheres with radiographic abnormalities including temporal sclerosis or previous neurosurgery were excluded from analysis. A kurtosis algorithm (with a maximum amplitude threshold of 4) was used to exclude abnormal events and interictal activity. One subject (S8) was excluded from the neural analyses due to excessive noise. Our hippocampal formation electrode coverage included contacts in the anterior hippocampus (AH; $n = 40$), posterior hippocampus (PH; $n = 26$), mid-hippocampus ($n = 8$), and entorhinal cortex (EC; $n = 12$).

## Anatomical localization

Participants had intracranial depth electrodes implanted at locations specified by the neurology team. Electrodes were laterally inserted into the specified regions with robotic assistance. Final electrode localization in the hippocampal formation subregions was determined by post-operative expert neuroradiology review.

## Power analyses

For the memory-evoked power analysis, spectral power was extracted using a continuous Morlet wavelet transform (wave number = 6) across 64 logarithmically spaced frequencies from 2 to 32 Hz. The analysis included all 1800-ms encoding periods following word onset, and 3000-ms buffers to avoid edge artifacts. Power measures at each session, electrode, and frequency were normalized relative to the mean and standard deviation of the baseline period 500 ms before

word onset. To establish significant differences between the conditions, a LME model was implemented using time-averaged power measures during encoding and setting condition (scopolamine/placebo) as fixed effects and electrodes as random effects. We used this LME framework because it accounts for intersubject differences in power measures, which are common for human hippocampal theta[99]. For the broadband power analysis, we extracted spectral power across 2–128 Hz and parameterized the aperiodic activity of the PSD of each electrode[57]. This method involves recursively fitting the aperiodic background curve (1/f) to the observed smoothed spectral signal, resulting in two parameters: an offset (or intercept) and a slope (or exponent). The offset parameter quantifies the magnitude of broadband activity. We assessed statistically significant differences in broadband power in two ways. First, we computed the distribution of the aperiodic offsets for all electrodes in each condition. Second, for each electrode and at each frequency we computed the difference of the PSD between the two conditions using Wilcoxon rank-sum tests.

## Phase reset analysis

In order to ensure phase values reflected true oscillations, we first implemented an established oscillation-detection routine ("Better Oscillation Detection" method, or BOSC)[61], which detects both sustained and transient oscillatory activity using thresholds for power and duration. Next, we extracted phase measures using a continuous Morlet wavelet transformation (wave number = 6) across 64 logarithmically spaced frequencies from 2 to 32 Hz. We computed the phase reset of oscillations at word onset using the ITPC measure[62,63] as follows:

$$ITPC(f,t) = \frac{1}{N} \left| \sum_{k=1}^{N} e^{i\varphi_k(f,t)} \right| \qquad (1)$$

where $f$ is the frequency for a given time $t$, $N$ is the number of trials, and and $e^{i\varphi_k}$ is the polar representation of the phase angle $\varphi$. This measure was computed separately for individual electrodes, and averaged at the subject-level for the group analysis. To assess significant differences in ITPC between conditions, we used an electrode-level nonparametric cluster permutation-based approach to correct for multiple comparisons across time and frequency[100]. To avoid spurious phase measures, we only computed the ITPC for electrodes that showed spectral peaks in the theta frequency range (2–10 Hz)[57]. Since ITPC calculations are highly susceptible to the number of observations, we excluded one subject (S7) who had fewer than sixty total trials in the scopolamine condition. For correlation plots, we measured phase reset in all conditions by computing the average ITPC in the 0–300-ms period, which is an interval where phase resetting was previously observed in the human hippocampus[41,42].

## Phase synchrony analysis

To assess connectivity within the hippocampal formation, we computed the phase-locking value (PLV) for all possible electrode pairs within a subject for each condition[67]. The PLV is a continuous measure between 0 and 1 that indicates how synchronized two electrodes are across trials. A low PLV indicates that phase differences vary significantly between the electrodes, whereas a high PLV indicates that the electrodes are highly synchronized across trials. The PLV is computed as follows:

$$PLV(t) = \frac{1}{N} \left| \sum_{k=1}^{N} e^{i(\phi_1(t,k) - \phi_2(t,k))} \right| \qquad (2)$$

where $\phi_1(t,k) - \phi_2(t,k)$ is the phase difference between electrodes. Significant differences between conditions were assessed using Wilcoxon rank-sum tests of the time-averaged PLVs for each electrode pair. PLV measures were calculated over the entire time course of encoding, based on methods employed in several previous

publications[44–47] and based on our observation that scopolamine-related synchrony changes were not locked to item presentation.

## Event-related potential analysis

To test whether phase reset measures were driven by event-related potentials (ERPs), we assessed whether changes in ERP significantly matched changes in power (1–10 Hz). Significant differences between conditions were assessed using Wilcoxon rank-sum tests at each timepoint −500 ms prior to and 1500 ms following encoding cue for each electrode. To compute the correlation between ERP and power changes, we used the timepoint of maximal ERP difference between conditions (± 150 ms) and its corresponding power measures in time.

## Gene expression analysis

The gene expression analysis included in the supplementary materials uses a previously published hippocampal single nucleus RNA sequencing (snRNAseq) dataset of 131,325 nuclei[86]. Tissue was donated by five patients who were undergoing temporal lobectomy to treat drug-resistant epilepsy. Anterior and posterior hippocampal tissue samples were flash frozen in liquid nitrogen within 20 min following resection. Nuclei were isolated from frozen hippocampal samples[101], and libraries were constructed using the droplet-based 10x Genomics platform (Chromium Single Cell 3' v2 or v3; #120237, #1000153). An Illumina NovaSeq 6000 was used for library sequencing. Sequences were aligned and mapped using CellRanger software (v.3.0.2, 10 × Genomics). Cell clustering, cell type identification, and gene expression analyses were executed with the Seurat R package and custom scripts. Normalized and integrated datasets were scaled by regressing for age, sex, epilepsy duration, batch, version of 10× chemistry, percent mitochondrial transcripts, and UMI number. Only cells containing >300 genes, <10000 UMIs, and <5% mitochondrial transcripts were retained for our analyses. After principal component and JackStraw analyses, principal components 1 through 25 and a resolution of 0.6 were used for Louvain clustering and UMAP.

## Reporting summary

Further information on research design is available in the Nature Portfolio Reporting Summary linked to this article.

## Data availability

The data that support the findings of this study are provided in the Source Data file. Source data are provided with this paper.

## Code availability

The task was coded using the publicly available Python Experiment-Programming Library (PyEPL; https://pyepl.sourceforge.net/). All data analyses were performed in MATLAB (R2018a) or Python (3.6), using publicly available software for behavioral analysis of free recall data (https://github.com/pennmem/pybeh) and human electrophysiology data (https://github.com/pennmem/ptsa and https://github.com/pennmem/cmlreaders). Other custom code is available at https://github.com/tgedankien/scopolamine. Gene sequences were aligned and mapped using CellRanger software (v.3.0.2, 10× Genomics).

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

## Acknowledgements

We are grateful to the patients for participating in our study. We thank Ehren Newman, Erfan Zabeh, Lukas Kunz, Thomas Donoghue, and Genevieve Konopka for valuable feedback. We thank Joseph Rudoler for providing technical support.

This project received support from National Institutes of Health grants R01-MH104606 (J.J.), R01-NS125250 (B.L.), and R01-NS107357 (B.L.).

## Author contributions

B.L. conceived the study and performed surgical procedures; D.M. assisted with patient screening and drug administration; R.T. and B.L. conducted the experiments; T.G., S.E.Q., J.J., and B.L. conceived the data analyses; T.G. and R.T. implemented the electrophysiological data analyses; H.M. conducted the gene expression analysis; T.G., J.J., and B.L. wrote the manuscript; J.J. and B.L. jointly supervised the project.

## Competing interests

The authors declare no competing interests.
