## [Peer Review File · Nature Communications]

Acetylcholine modulates the temporal dynamics of human theta oscillations during memoryREVIEWER COMMENTS

Reviewer #1 (Remarks to the Author):

In this manuscript, Gedankian and colleagues examine the effects of cholinergic blockade on theta oscillatory activity in the human hippocampus. Using scopolamine administered to patients with intracranial electrodes, the authors investigate changes in theta oscillatory power, phase, and synchrony as the participants are performing a verbal recall task. This is an important topic, and the paper addresses some key questions regarding the role of the cholinergic system in memory. Previous studies have suggested that memory impairment is related to cholinergic dysfunction, but the mechanism through which cholinergic dysfunction or blockade disrupts memory is unclear. Given the known role of theta oscillations in the medial temporal lobe in human episodic memory, the authors specifically examine how administration of scopolamine affects theta activity in these structures. They find that scopolamine decreases memory performance, decreases narrowband 2-4 Hz theta band power, reduces theta inter trial phase consistency, and reduces theta synchronization in the hippocampus. This is a well written and clear study, and there is a lot of appreciate and like about the manuscript. There are some suggestions that would potentially help improve the overall conclusions that can be drawn from the work.

The authors show that there is a clear decrease in 2-4 Hz theta for scopolamine, localized to the posterior hippocampus. But when analyzing broadband power, it appears that the majority of electrodes actually show an increase in 2-4 hz power in scopolamine compared to placebo (fig 2G). Why is this different? This would suggest that the decrease in power seen with scopolamine is actually a relative decrease compared to the overall elevated power (e.g., normalized to the new baseline). Which would then suggest that perhaps the main effect of scopolamine is not necessarily a decrease in theta power, but instead is that with cholinergic blockade one is now unable to get elevated theta power during memory encoding that you would see without blockade given the new baseline (e.g., fig 2B)?

On a related note, how does this change in theta power activity relate to previous reports on the role of theta in subsequent memory (the subsequent memory effect). In prior work, there has been evidence that theta power actually decreases in correct compared to incorrect trials. Here, it looks like the authors just examine overall power during encoding but what about difference in power between correct and incorrect trials?

For the analysis of ITPC, there are two things to note. First, the difference in the example in Fig 3b appears to be localized to 8hz, not the 2-4 hz band described in figure 2. The authors examine ITPC separately in the 2-4 Hz band and in the 4-10 Hz band as shown in this example. How should one interpret the changes in ITPC in the faster band in the absence of changes in power. Even across subjects, the relation between itpc disruption and memory performance is primarily in the 4-10 Hz band,

yet the power changes are in lower frequencies. How should this be reconciled? The group level ITPC analysis demonstrates much larger regions of difference between drug and control. This would suggest a general disruption from scopolamine within the theta band, so I wonder how to interpret the data and whether we should be focusing on specific sub bands of theta, or general disruptions to the entire low frequency range.

Second, in the example electrode in Figure 3b, is this an EC electrode? Why not instead present an electrode from the hippocampus since scopolamine's effects on theta power were limited to posterior hippocampus? Moreover, the PSD for this example electrode shows no difference between scopolamine and placebo.

In addition, for the correlations in Fig 4, are these just hippocampal electrodes? Because it appears that for one subject (Fig 4c) there is an EC electrode that is used for the correlation presented in Fig 4b?

For the analysis of hippocampal phase synchrony, there are again different effects in different frequency bands. This may certainly be due to variability between subjects, but is this similar variability in the changes in power. In other words, are the differences in frequencies across subjects that are relevant for phase synchrony mirrored by differences in frequencies that are relevant for power across subjects? In the discussion, the authors say that these differences just implicate all of theta band, but if that were the case, would it make more sense to conduct the analyses as such. This also raises a question about the conclusion that scopolamine effects are limited to type 1 or type 2 theta.

The authors also claim is that more synchrony between hippocampal electrodes is present in placebo compared to scopolamine condition. How is this compared? In Fig 5d, both show increases compared to baseline. The authors use a two proportion z-test, but is this significant for every frequency? Are there questions about multiple frequencies and therefore multiple comparisons?

Over what time window is the PLV measured? Is it over the whole encoding trial? It appears that for the ITPC analysis, they picked point of maximum reset and defined a 300 ms window around that point. What is the window of analysis here, and is it similar?

The data in 5A suggest that perhaps one effect of scopolamine is that it just slows down theta cycle, leading to less PLV over the course of a trial. It would be interesting to know if this indeed happens and if the authors could provide some data investigating this question.

Reviewer #2 (Remarks to the Author):

Gedankien et al. investigated the effects of the muscarinic blocker scopolamine on human epilepsy patients with intracranial EEG electrodes, performing a verbal episodic memory task. Scopolamine impaired episodic memory but spared the performance of a simple arithmetic task (although with slowed responses). First, they showed that increased slow theta activity during the task was lower under scopolamine, while there was an increase in broad-band/aperiodic spectral power. Next, the Authors showed that scopolamine also disrupted stimulus-induced phase reset of ongoing theta oscillation, which predicted the extent of memory impairment (interestingly, stronger in the fast than the slow theta band). Finally, there was an increase in inter-electrode phase synchrony in the hippocampus in placebo-treated patients during the task, which was also disrupted by scopolamine.

It is an important question how acetylcholine controls human memory formation, and this thorough study represents a conceptual advance based on hard-to-obtain high-quality human data, using state-of-art data analysis techniques. I have a few suggestions and comments.

1. 'Phase reset differences used in all correlations were based on measurements at the timepoint of the maximal phase reset difference between conditions (± 150 ms).' Doesn't this introduce a statistical bias? What would happen at constant time? Relatedly, in Fig. S6 it is somewhat surprising that all electrodes reset to different phase angles without apparent regularities. Can this have anything to do with measuring these at different time lags?

2. Much of the cholinergic action on hippocampus may be indirect through MS GABAergic neurons (see e.g. Dannenberg et al., "Synergy of Direct and Indirect Cholinergic Septo-Hippocampal Pathways Coordinates Firing in Hippocampal Networks" and Yang et. al. "The menagerie of the basal forebrain: how many (neural) species are there, what do they look like, how do they behave and who talks to whom?"). Since systematic cholinergic blockade cannot differentiate between direct and indirect effects, it may be worth it to note this as a discussion point. Relatedly, possible off-target (outside hippocampus) and side effects would deserve somewhat more discussion.

3. The 'aperiodic' power increase could use a bit more explanation: it may look confusing at first, that there's a reduction in the lower theta band, but an aspecific broad-band increase, which also includes this band, while this is not visible in Figure 2B (I am assuming it is not time-locked, so it may be removed by the normalization procedure). Also, it could be highlighted that the 'aperiodic' activity is traditionally characterized by a linear fit on the log-log power spectrum, and a shift in this fit is what we are looking for (as in Figure 2E).

4. Muscarinic blockers also block pre-synaptic auto-receptors and thus generally elevate ACh levels. To capitalize on this, they (primarily atropine, but also others) were used to enhance SNR in early ACh measurements (the 'cortical cup' method; see e.g. Phillis: "Acetylcholine release from the central nervous system: a 50-year retrospective", 2005). This may be OK, as it probably also blocks most postsynaptic muscarinic receptors, while nicotinic receptors tend to desensitize quickly, and scopolamine is indeed widely interpreted as a 'blocker'. Nevertheless, it would be nice to see the Authors' take on this (maybe even as a discussion point).

5. Minor points.

Please explain Fig.1G.

Fig.2C. 'Contrast denotes t-statistics from linear mixed effect (LME) models computed at every frequency and timepoint.' – I don't really see it.

Fig.3B. Why '125 ms following encoding cue for placebo'? Was this the peak location in the ITPC?

Fig.3C – Why is there no 'broad-band power increase' in this graph as compared Fig.2E?

Fig.5C. Shouldn't 'mean PLV over time' rather read 'mean PLV over frequency'?

I saw 'an hippocampal' instead of 'a hippocampal' at a few places.

Phase reset analysis. In the definition in line 388, it should read ϕ^k instead of ϕ . If I'm correct, k is an index and not an exponent, so ϕ_k would probably be better than ϕ^k .

'A low PLV means that the electrode pair is completely unsynchronized' Sure? The measure seems to be built on pairwise phase differences. In the formula (equation 2), is j the imaginary number? If so, it would be better to use i to be consistent with equation 1.

Fig.S7A, right panel is not fully clear. What is the shading here?

Will the data be available?

I believe that openness and transparency can increase the fairness of peer review. Therefore, I decided to sign my reviews.

Point-by-point response to reviewer comments

Below are detailed responses and descriptions of changes made following the reviewers' comments and suggestions. Black text indicates the reviewer comment, followed by our response in blue. In the revised manuscript PDF file, for reviewers' convenience, blue coloring indicates text that we changed in response to a reviewer suggestion.

Response to Reviewer 1

In this manuscript, Gedankian and colleagues examine the effects of cholinergic blockade on theta oscillatory activity in the human hippocampus. Using scopolamine administered to patients with intracranial electrodes, the authors investigate changes in theta oscillatory power, phase, and synchrony as the participants are performing a verbal recall task. This is an important topic, and the paper addresses some key questions regarding the role of the cholinergic system in memory. Previous studies have suggested that memory impairment is related to cholinergic dysfunction, but the mechanism through which cholinergic dysfunction or blockade disrupts memory is unclear. Given the known role of theta oscillations in the medial temporal lobe in human episodic memory, the authors specifically examine how administration of scopolamine affects theta activity in these structures. They find that scopolamine decreases memory performance, decreases narrowband 2-4 Hz theta band power, reduces theta inter trial phase consistency, and reduces theta synchronization in the hippocampus. This is a well written and clear study, and there is a lot of appreciate and like about the manuscript. There are some suggestions that would potentially help improve the overall conclusions that can be drawn from the work.

We thank the reviewer for these comments.

The authors show that there is a clear decrease in 2-4 Hz theta for scopolamine, localized to the posterior hippocampus. But when analyzing broadband power, it appears that the majority of electrodes actually show an increase in 2-4 hz power in scopolamine compared to placebo (fig 2G). Why is this different? This would suggest that the decrease in power seen with scopolamine is actually a relative decrease compared to the overall elevated power (e.g., normalized to the new baseline). Which would then suggest that perhaps the main effect of scopolamine is not necessarily a decrease in theta power, but instead is that with cholinergic blockade one is now unable to get elevated theta power during memory encoding that you would see without blockade given the new baseline (e.g., fig 2B)?

The reviewer's description of our results is correct. We find that scopolamine injection leads to two different effects: 1) greater broadband power and 2) lower amplitude memory-evoked narrowband theta oscillations. The revised paper more clearly distinguishes these effects, by now using the terms "memory-evoked" and "narrowband" to describe the 2–4 Hz effects. We also clarified the description and distinction between the interpretations of broadband and narrowband oscillations in the Results, page 6, and Methods, page 13.

On a related note, how does this change in theta power activity relate to previous reports on the role of theta in subsequent memory (the subsequent memory effect). In prior work, there has been evidence that theta power actually decreases in correct compared to incorrect trials. Here, it looks like the authors just examine overall power during encoding but what about the difference in power between correct and incorrect trials?

We appreciate this question about the effect of scopolamine on the subsequent memory effect (SME) and it is in fact something we had analyzed previously. However, we had found that the scarce number

of successfully recalled events in the scopolamine condition limited statistical power to measure an interaction between SME effects and drug administration.

Nonetheless, to address the reviewer's question, we added to the manuscript new analyses of SME effects. The paper now reports an analysis of SME power changes in the placebo condition, where we found a trend towards greater power for successful versus unsuccessful encoding (**new Fig. S5**). Importantly, this effect appears at the same band as the one that showed decreases for scopolamine. As we explain in the paper, this result, which is consistent with earlier studies (Lega et al., 2012; Staudigl and Hanslmayr, 2013; Kota et al., 2020), helps us better interpret our results because they suggest that our scopolamine-related theta effects reflect a modulation of the same memory-related theta oscillations that were previously reported as being linked to successful memory formation.

For the analysis of ITPC, there are two things to note. First, the difference in the example in Fig 3b appears to be localized to 8hz, not the 2-4 hz band described in figure 2. The authors examine ITPC separately in the 2-4 Hz band and in the 4-10 Hz band as shown in this example. How should one interpret the changes in ITPC in the faster band in the absence of changes in power. Even across subjects, the relation between itpc disruption and memory performance is primarily in the 4-10 Hz band, yet the power changes are in lower frequencies. How should this be reconciled? The group level ITPC analysis demonstrates much larger regions of difference between drug and control. This would suggest a general disruption from scopolamine within the theta band, so I wonder how to interpret the data and whether we should be focusing on specific sub bands of theta, or general disruptions to the entire low frequency range.

The reviewer highlights important considerations regarding the distinction between slow and fast theta oscillations, and how that influences the interpretations of the power and ITPC effects. Our analysis framework was motivated by the “two theta” model of hippocampal physiology, whereby studies in animals have shown that cholinergic drugs have differential effects on theta sub-bands—mainly causing stronger power disruptions to slow (“Type 2”) theta rather than fast (“Type 1”) theta (Kramis et al., 1975; Dunn et al., 2021). Our findings support this hypothesis in general, insofar as we observed power changes from scopolamine that were strongest in the slower theta band.

However, as the reviewer notes, our observed ITPC effects spanned the entire frequency range of theta. We believe these results are consistent with the view that the human hippocampal region possesses multiple distinct theta oscillators. These individual theta generators, each with their own intrinsic frequency, may interact with one another to support cognition, and be affected by scopolamine both directly and indirectly—thus, leading to observations of ITPC effects across the full theta range. This model is consistent with earlier findings of theta often being spread across a range of frequencies in humans (Goyal et al., 2020) and in animals (Goutagny et al., 2009). With this model in mind, because there is coupling between these distinct theta oscillators, we believe that the observed changes in fast theta ITPC could be driven indirectly by power effects in slow theta oscillators and/or by a selective effect of scopolamine on fast theta oscillators.

In the revised manuscript, we have now expanded our discussion of these ideas by further describing how the potential coupling between multiple distinct theta oscillators informs the interpretation of our results and their relation to prior literature related to animal Type 1 & Type 2 theta oscillations (**page 10**).

Second, in the example electrode in Figure 3b, is this an EC electrode? Why not instead present an electrode from the hippocampus since scopolamine's effects on theta power were limited to posterior

hippocampus? Moreover, the PSD for this example electrode shows no difference between scopolamine and placebo.

While our study included electrodes across both the hippocampus and EC, where we saw similar effects, we agree that having our first key example from the EC was confusing. To enhance clarity, we changed this figure to a hippocampal electrode and provided additional single electrode examples from the EC and hippocampus in a new supplemental figure (**new Fig. S8**). We also removed the PSD plot from the figure to avoid confusion with the power results.

In addition, for the correlations in Fig 4, are these just hippocampal electrodes? Because it appears that for one subject (Fig 4c) there is an EC electrode that is used for the correlation presented in Fig 4b?

As mentioned above, the study included both the hippocampus and EC regions. In Figure 4C, each dot reflects the mean ITPC across all electrodes for one subject – across both hippocampal and EC contacts, depending on the subject's specific coverage. To clarify this issue, the paper now more fully describes how we aggregated data across electrodes in the Results section **page 4**.

For the analysis of hippocampal phase synchrony, there are again different effects in different frequency bands. This may certainly be due to variability between subjects, but is this similar variability in the changes in power. In other words, are the differences in frequencies across subjects that are relevant for phase synchrony mirrored by differences in frequencies that are relevant for power across subjects? In the discussion, the authors say that these differences just implicate all of theta band, but if that were the case, would it make more sense to conduct the analyses as such. This also raises a question about the conclusion that scopolamine effects are limited to type 1 or type 2 theta.

The reviewer makes an insightful point analogous to the one above about the distinction between slow and fast theta effects in the power and ITPC analyses. As above, we believe our observed synchrony effects can be explained by the interplay between distinct slow and fast theta oscillators, which dictate the composition of a subject's ongoing theta oscillations (Goyal et al., 2020). Some subjects predominantly show narrowband theta while others show effects across a range of theta frequencies, thus leading to observed changes in the broad theta range in situations where these oscillations are coupled.

Additionally, in a supplementary analysis (**Fig. S14**), we did not find significant correlations between changes in power and changes in phase synchrony. These results support the idea that power and phase synchrony reflect distinct processes that are not necessarily mirrored on one another, and likely involve a more complex interaction between local and distal slow and fast theta generators.

We have now updated the manuscript to more fully discuss our interpretations related to slow versus fast theta oscillations in the context of our synchrony effects (**page 10**).

The authors also claim is that more synchrony between hippocampal electrodes is present in placebo compared to scopolamine condition. How is this compared? In Fig 5d, both show increases compared to baseline. The authors use a two proportion z-test, but is this significant for every frequency? Are there questions about multiple frequencies and therefore multiple comparisons?

We have revised the paper to more clearly describe our methods for analyzing synchrony changes across electrodes and frequencies. For the analysis in Figure 5D, the text now explains that we compute a single statistic across the entire frequency band, by first measuring the average proportions of significant electrodes across all frequencies in the band and then performing a single z-test. Thus, because only a single statistic is calculated, there is no concern about multiple comparisons. We have

clarified this point in the Methods section, **page 14**, and in the figure caption. To stay consistent with the analysis structure used for power and ITPC, we have also included group-level statistics separately for the slow (2–4 Hz) and fast (4–10 Hz) theta bands and displayed it in a bar plot in **new Figure 5E**.

Over what time window is the PLV measured? Is it over the whole encoding trial? It appears that for the ITPC analysis, they picked point of maximum reset and defined a 300 ms window around that point. What is the window of analysis here, and is it similar?

In the revised paper we now more fully explain that we calculated PLV over the entire time course of encoding, which is different from our analysis of phase reset (ITPC), which was focused at a specific time point relative to item presentation. The reason we average PLV over an extended time period is because the synchrony effects were extended in time (see Figure 5D) and because this method of examining synchrony over extended time intervals followed those employed in several previous publications (Eichenbaum, 2000; Solomon et al., 2017; Gruber et al., 2018; Choi et al., 2020).

The data in 5A suggest that perhaps one effect of scopolamine is that it just slows down theta cycle, leading to less PLV over the course of a trial. It would be interesting to know if this indeed happens and if the authors could provide some data investigating this question.

We recognize the example in Figure 5A gave the indication that scopolamine might slow down theta oscillations following item presentation. To test whether this theta slowing was robust, we ran a new analysis comparing synchrony for early versus late time periods of encoding, but we did not find significant differences (p 's > 0.05, paired t-tests). We now mention this finding on **page 8** in the Results section, and we have changed the example in Figure 5A to include a different trace that no longer gives the appearance that there was a theta frequency decrease across trials.

Response to Reviewer 2

Gedankien et al. investigated the effects of the muscarinic blocker scopolamine on human epilepsy patients with intracranial EEG electrodes, performing a verbal episodic memory task. Scopolamine impaired episodic memory but spared the performance of a simple arithmetic task (although with slowed responses). First, they showed that increased slow theta activity during the task was lower under scopolamine, while there was an increase in broad-band/aperiodic spectral power. Next, the Authors showed that scopolamine also disrupted stimulus-induced phase reset of ongoing theta oscillation, which predicted the extent of memory impairment (interestingly, stronger in the fast than the slow theta band). Finally, there was an increase in inter-electrode phase synchrony in the hippocampus in placebo-treated patients during the task, which was also disrupted by scopolamine.

It is an important question how acetylcholine controls human memory formation, and this thorough study represents a conceptual advance based on hard-to-obtain high-quality human data, using state-of-art data analysis techniques. I have a few suggestions and comments.

Thank you for these positive comments about our work!

1. 'Phase reset differences used in all correlations were based on measurements at the timepoint of the maximal phase reset difference between conditions (± 150 ms).' Doesn't this introduce a statistical bias? What would happen at constant time?

There is no statistical bias because we measured phase reset differences at a single time window (0–300 ms) that was fixed across all subjects, which we had preselected based on earlier studies (Lin et al., 2017; Kota et al., 2020). We apologize for the poorly worded text that is quoted, and we have

revised it accordingly. In the revised paper we now do a better job describing this method, by stating that this 0–300 ms window was chosen a priori because it would include the period with peak phase reset—as it was also confirmed by the cluster-based permutation test (Maris and Oostenveld, 2007) that corrected for multiple comparisons across time and frequency in Figure 3C. The revised text in the Methods section (**page 14**) now reads: “For correlation plots, we measured phase reset in all conditions by computing the average ITPC in the 0–300-ms period, which is an interval where phase resetting was previously observed in the human hippocampus (Lin et al., 2017; Kota et al., 2020)”.

Relatedly, in Fig. S6 it is somewhat surprising that all electrodes reset to different phase angles without apparent regularities. Can this have anything to do with measuring these at different time lags?

The reviewer also notes an interesting feature of our data, which is that even in the placebo condition, the specific phase to which theta oscillations are resetting (aligning) is not uniform across electrodes. We also found that the phase differences did not seem related to variations in time lag ($p > 0.05$, circular-linear correlation). Our findings are consistent with previously published work using single unit recordings from the human MTL, in which there was no uniform phase alignment among memory-sensitive neurons (Yoo et al., 2021).

We have added a sentence to the Results section (**page 6**) mentioning these findings. Investigating the fundamental relevance of variations in the angle of oscillatory phase resets is an interesting topic for future work, in which researchers may probe how the specific angle of phase resets changes with behavior as well as with intersubject variations in anatomy and physiology.

2. Much of the cholinergic action on hippocampus may be indirect through MS GABAergic neurons (see e.g. Dannenberg et al., “Synergy of Direct and Indirect Cholinergic Septo-Hippocampal Pathways Coordinates Firing in Hippocampal Networks” and Yang et. al. “The menagerie of the basal forebrain: how many (neural) species are there, what do they look like, how do they behave and who talks to whom?”). Since systematic cholinergic blockade cannot differentiate between direct and indirect effects, it may be worth it to note this as a discussion point. Relatedly, possible off-target (outside hippocampus) and side effects would deserve somewhat more discussion.

We thank the reviewer for this helpful suggestion, as it speaks directly to potential mechanisms by which cholinergic blockade may influence hippocampal theta oscillations. To explore this issue further, we conducted a new analysis examining gene expression in the human hippocampus. This analysis shows evidence for receptors that mediate cholinergic effects in the human hippocampus.

In brief, we reanalyzed a previously published dataset (Ayhan et al., 2021) and found robust expression of CHRM3 receptor RNA in human hippocampal tissue samples from a similar population of surgical epilepsy patients (**new Fig. S16**). We believe these findings offer preliminary support for the existence of a direct septo–hippocampal cholinergic pathway that could potentially modulate theta, consistent with the model articulated in Dannenberg et al. 2015. In addition to the new supplementary material (**new Fig. S16, new Methods, and new Supplementary Discussion**), these findings and interpretations are mentioned in a new paragraph discussing the potential direct and indirect mechanisms of cholinergic blockade in the Discussion section, **page 11**. We believe this extended analysis will enhance the interest that our work garners among scientists interested in cholinergic mechanisms and can serve as the basis of further experimentation using in vitro preparations of human tissue.

A related but distinct concern brought by the reviewer is the possibility that the effects of scopolamine in the hippocampal area result from *off-target* or *side effects* of scopolamine in the neocortex. Although we believe the mnemonic effects of scopolamine reflect its impact on hippocampal physiology in our

experimental paradigm (which involves a hippocampus-dependent task), there is also substantial evidence for cholinergic modulation of non-memory-related behaviors (Himmelheber et al., 2000; Sarter et al., 2005; Klinkenberg et al., 2011; Sviatko and Hangya, 2017; Teles-Grilo Ruivo et al., 2017). For that reason, we agree with the reviewer that it is important to discuss the possibility that cholinergic processes on the hippocampus occur via indirect pathways from the cortex, and we have now expanded our discussion on this topic on **page 12**.

3. The 'aperiodic' power increase could use a bit more explanation: it may look confusing at first, that there's a reduction in the lower theta band, but a specific broad-band increase, which also includes this band, while this is not visible in Figure 2B (I am assuming it is not time-locked, so it may be removed by the normalization procedure). Also, it could be highlighted that the 'aperiodic' activity is traditionally characterized by a linear fit on the log-log power spectrum, and a shift in this fit is what we are looking for (as in Figure 2E).

As suggested, we have added a more detailed explanation of the distinction between aperiodic and periodic (narrowband) signals in the Results section on **page 6**. As we explain there, we believe these two different effects are important because they reflect different underlying physiological phenomena, with broadband signals reflecting overall changes in E/I balance and narrowband signals reflecting rhythmic oscillations. Also, as suggested, we have added more details on how we measured the aperiodic activity in the Methods, **page 13**. We also note that our interpretation that scopolamine modulates broadband power, which correlates with E/I balance, is also consistent with models like the ones cited by the reviewer in the previous point (Dannenberg et al., 2015; Yang et al., 2017).

4. Muscarinic blockers also block pre-synaptic auto-receptors and thus generally elevate ACh levels. To capitalize on this, they (primarily atropine, but also others) were used to enhance SNR in early ACh measurements (the 'cortical cup' method; see e.g. Phillis: "Acetylcholine release from the central nervous system: a 50-year retrospective", 2005). This may be OK, as it probably also blocks most postsynaptic muscarinic receptors, while nicotinic receptors tend to desensitize quickly, and scopolamine is indeed widely interpreted as a 'blocker'. Nevertheless, it would be nice to see the Authors' take on this (maybe even as a discussion point).

The most widely accepted interpretation of the effects of scopolamine in the brain is a blockade of postsynaptic muscarinic receptors. However, as pointed out by the reviewer, a competing mechanism is a blockade of presynaptic muscarinic auto-receptors, which could lead to elevated release of acetylcholine in the brain. However, because prior work suggests that the primary effect driving memory impairment is the blockade of muscarinic receptors (Hasselmo, 2006), we believe that our observations most likely reflect the effects of postsynaptic blockade of muscarinic receptors.

We have added a discussion of issue to the Discussion section on **page 11**, in which we acknowledge that additional in vitro work on human tissue samples is likely necessary to confirm that the effects of scopolamine in the human hippocampus rely overall on postsynaptic blockade of muscarinic receptors rather than an increase in synaptic ACh due to autoreceptor blockade. We also link these ideas to the results of our new analysis of gene expression, which demonstrates that CHRM receptors are present in key cell populations in the hippocampus that provide a plausible mechanism for postsynaptic effects of scopolamine (**see new Fig. S16 and new Supplementary Discussion**). We have added the useful citation to the Phillis 2005 review, which highlights how long scientists have been interested in cholinergic modulation of the hippocampus.

5. Minor points.

Please explain Fig.1G.

Figure 1G displays the percentage of electrodes showing significance increases in power for each condition at each frequency. We now have clarified the description of methods and interpretation of Figure 1G in its caption (page 5) and the Results (page 6).

Fig.2C. 'Contrast denotes t-statistics from linear mixed effect (LME) models computed at every frequency and timepoint.' – I don't really see it.

To more clearly illustrate the decrease in encoding-related theta power following scopolamine, we have changed Figure 2C to a line plot showing the mean normalized 2–4 Hz power per condition.

Fig.3B. Why '125 ms following encoding cue for placebo'? Was this the peak location in the ITPC?

We have edited the caption of Figure 3B to indicate that the labeled time point corresponds to the peak in ITPC between 0–300ms for the placebo condition.

Fig.3C – Why is there no 'broad-band power increase' in this graph as compared Fig.2E?

We apologize for this confusion. Most electrodes show a broadband power increase as in Figure 2E, but the particular example in Figure 3C was unusual and did not show this broadband effect. Because it is not representative of most of our data, we have now removed it from the paper.

Fig.5C. Shouldn't 'mean PLV over time' rather read 'mean PLV over frequency'?

We have edited this text to now read "mean PLV as a function of frequency."

I saw 'an hippocampal' instead of 'a hippocampal' at a few places.

Thank you, we have fixed these typos.

Phase reset analysis. In the definition in line 388, it should read ϕ^k instead of ϕ . If I'm correct, k is an index and not an exponent, so ϕ_k would probably be better than ϕ^k .

We thank the reviewer for pointing out this issue and we have changed ϕ^k to ϕ_k (page 14).

'A low PLV means that the electrode pair is completely unsynchronized' Sure? The measure seems to be built on pairwise phase differences. In the formula (equation 2), is j the imaginary number? If so, it would be better to use i to be consistent with equation 1.

We rephrased this to state that a PLV of zero would reflect no synchrony and we have reworded the description in the Methods section page 14 to explain this idea, using a description consistent with Lachaux et al. 1999. We also thank the reviewer for pointing out the inconsistency with the imaginary number notation in the PLV equation. As suggested, we changed j to i to stay consistent with equation 1 (page 14).

Fig.S7A, right panel is not fully clear. What is the shading here?

The revised caption of this figure (now Fig. S10A) explains that the gray shading in the right panels denotes areas of non-significant effects. If the line extends outside the gray regions, then it is

significant. Note that this scheme is consistent with Figures 10B. We added additional text to the caption to more fully explain that a consistent coloring scheme is used across Figures S10A and 10B.

Will the data be available?

Yes. The manuscript now states that the data that support the findings of this study are available upon request from the corresponding author (B.L.).

REVIEWERS' COMMENTS

Reviewer #1 (Remarks to the Author):

The authors have sufficiently addressed the comments raised in the first round of review. I commend the authors on their work and on their revision, and this manuscript will be a valuable addition to the literature.

Reviewer #2 (Remarks to the Author):

In the supplementary gene expression analysis, the Authors say 'We believe these findings directly support the existence of an indirect pathway...' I thought the Authors argue for the direct pathway here (or I may be misreading this somehow). Nevertheless, this is a really minor aside and the Authors adequately answered all of my points. I think this study makes a significant contribution to understanding human theta oscillations.

1. REVIEWERS' COMMENTS

Reviewer #1 (Remarks to the Author):

The authors have sufficiently addressed the comments raised in the first round of review. I commend the authors on their work and on their revision, and this manuscript will be a valuable addition to the literature

We thank Reviewer #1 for their feedback.

Reviewer #2 (Remarks to the Author):

In the supplementary gene expression analysis, the Authors say 'We believe these findings directly support the existence of an indirect pathway...' I thought the Authors argue for the direct pathway here (or I may be misreading this somehow). Nevertheless, this is a really minor aside and the Authors adequately answered all of my points. I think this study makes a significant contribution to understanding human theta oscillations.

We apologise for the typo. We changed the text to "We believe these findings support the existence of an **direct** pathway [...]". We thank Reviewer #2 for his feedback.